# Strategic management and risk control of emergency hospital construction: SWOT and STPA framework from a systems thinking perspective

**Dongliang Zhu**[1,2]*, **Song Wang**[1,2], **Yaru Li**[1,2]

1 College of Architecture and Civil Engineering, Xinyang Normal University, Xinyang, China, 2 Henan New Environmentally-Friendly Civil Engineering Materials Engineering Research Center, Xinyang Normal University, Xinyang, China

* zdl18075343973@163.com

**Data Availability Statement:** All relevant data are within the paper and its Supporting information files.

## Abstract

The construction of emergency hospitals is crucial for ensuring medical service provision during disasters. Assembled buildings have emerged as the preferred choice for large-scale emergency hospitals due to their rapid construction and high quality. However, the construction of emergency hospitals involves the collaboration of multiple departments, and there is a lack of research on the management of such construction projects. Given the urgent need for emergency hospitals, analyzing potential hazards in the construction process from a systemic perspective is essential to manage their construction effectively. In this study, the SWOT and STPA methods are employed to investigate the construction management of emergency buildings, with the Wuhan Vulcan Mountain Hospital in China serving as a case study for emergency management analysis. This study can provide ideas for emergency hospital management and a basis for controlling possible emergency construction accidents.

## 1. Introduction

An emergency hospital is a facility that provides temporary medical care services for individuals affected by natural disasters or other emergencies. It is a crucial component of emergency medicine, encompassing medical care, education, research, rescue, and transportation. Constructing an emergency hospital requires efficient and rapid construction technology and management techniques, considering various factors such as medical functions, environmental adaptation, energy efficiency, and environmental protection [1]. As a result, managing the construction of emergency hospitals is a complex and challenging task. Throughout history, emergency hospitals have played a significant role in human development. In the 14th century, the plague emerged as a critical issue in Europe [2]. During the middle and late nineteenth century, European and American nations established medical services in response to work-related injuries, train accidents, and other unforeseen events [3]. In World War II, emergency tent hospitals were developed [4], and as technology advanced, ambulances, fire engines, and

**Funding:** There is no problem with the financial aspects of our slip-up, and the grantees are the corresponding and first authors of this paper. ZDL was supported by the Xinyang Normal University Graduate Research Innovation Fund (2022KYJJ089), which provided support for the publication of the paper and data collection.

**Competing interests:** The authors have declared that no competing interests exist.

other rapid emergency mobile equipment were swiftly produced. However, in cases where the magnitude of the catastrophe is significant and many people are affected, current mobile emergency equipment struggles to handle emergency tasks effectively. In such situations, emergency hospitals remain the preferred option for disaster mitigation [5].

Currently, assembled structures are the predominant way of constructing emergency buildings [6]. The assembled building has the following features, which make the assembled structure frequently utilized in emergency construction. Firstly, assembled structures allow for speedy construction [9]. Hospitals must be set up quickly in emergencies to address urgent medical needs. By utilizing prefabricated components and modular design, assembled buildings can be produced in factories and assembled on-site, significantly reducing construction time and improving efficiency [7]. Secondly, assembled buildings are known for their high quality. Emergency hospitals must meet strict medical and safety standards and fulfil the requirements for rescue and treatment [8]. By adopting standardized production and management, assembled buildings can enhance the precision and quality of components, minimize construction errors and defects, and ensure structural stability, functional completeness, comfort, and suitability [9]. Thirdly, the constructed building offers the advantage of energy conservation and environmental preservation [10]. Emergency hospitals must address environmental adaptation and sustainable development and minimize the environmental effects. Emergency hospitals must address environmental adaptation and sustainable development to minimize environmental impact and harm. The assembled building utilizes lightweight, demountable, low-cost, energy-saving, and recyclable materials and technologies. This approach can reduce construction and operation costs, minimize resource consumption and waste, and promote green construction and sustainable development of the emergency hospital [11].

Although there are numerous advantages to using assembled components for constructing emergency hospitals, there is still a need to investigate how to manage such projects' construction effectively, which is primarily due to the time-sensitive nature of emergency hospitals, which often require collaboration among multiple departments and rapid coordination [12]. Many scholars have conducted studies on emergency management. Several scholars have conducted studies on emergency management, considering various factors influencing it. These factors include the type, magnitude, severity, duration, frequency of occurrence, and other characteristics of the catastrophe, as they significantly impact emergency management's urgency, requirements, and complexity [12]. Catastrophe factors are crucial in emergency management as they form their basis. Different types of disasters have unique requirements and challenges. The severity of a sudden disaster, the environment, the affected population, and other factors determine emergency management's objectives, impacts, and primary concerns. Furthermore, research has demonstrated that emergency management involves multiple government agencies, non-profit organizations, businesses, and institutions. These entities have systems for coordinating efforts, rules, regulations, and norms [13, 14]. Building emergency hospitals is vital for emergency rescue. Hence, emergency construction management is fundamental to emergency management. However, there is a lack of research on emergency hospital construction management, as most academic studies focus on the macroscopic perspective of emergency management. In comparison, virtually little study has been undertaken on emergency construction management [15, 16].

This study aims to provide a new idea of emergency construction management to the rapid emergency assembly hospital to ensure the rapid construction of the emergency hospital. Firstly, SWOT analysis was used to analyze the internal and external environment of the emergency hospital construction to determine the advantages and disadvantages of the construction unit in the event of a disaster. Then, System Theoretic Process Analysis (STPA) was used to

analyze the construction of the emergency assembly hospital and to identify the "hazards" of the emergency assembly hospital during the construction process.

## 2. Literature review

This paper aims to address construction management issues in the construction of emergency buildings, so it focuses on research in emergency construction management and emergency hospitals.

### 2.1 Emergency management and emergency construction management

Emergency management refers to the activities that respond to sudden public incidents, including implementing effective prevention, preparation, response, and recovery measures to protect the safety of people's lives and properties and maintain social stability [17]. Emergency management is an interdisciplinary, cross-sector, and multi-level comprehensive discipline involving various aspects such as politics, economy, society, law, management, psychology, and technology. In recent years, with the frequent occurrence and expanding impact of sudden public incidents, emergency management research has received widespread attention and importance from academic and practical communities both domestically and internationally. Scholars have conducted institutional research on emergency management, such as the establishment and reform of the Federal Emergency Management Agency (FEMA) in the United States, the development and implementation of the National Response Framework (NRF) and the National Incident Management System (NIMS) [18]. Scholars have also explored emergency management decision-making, hoping to find effective response measures [17]. However, different risks require different emergency responses. Based on this, scholars have proposed risk assessment and early warning mechanisms for emergency management. Table 1 summarizes the main research perspectives and research deficiencies in emergency management. As shown in Table 1, these studies have made significant contributions to emergency management, but there are still deficiencies. These studies have improved the theoretical system of emergency management but cannot provide operational guidance for emergency rescue personnel, especially in the construction of emergency hospitals, where these studies cannot provide sufficient guidance.

Emergency construction management is a crucial component of emergency management, providing guidance on construction management in emergencies. However, emergency

**Table 1. Mainstream research perspectives on emergency management.**

| Object | Perspective | Methods | Deficiencies | Reference |
|---|---|---|---|---|
| Management system | Cognitive analytics management | Cognitive scorecard | Unable to control potential hazards | [19] |
| | Disaster message screening | Expert Interviews | Inability to take into account the diversity of the environment | [20] |
| | Emergency communications management | Optimization algorithms | Dependent on algorithmic accuracy and empirical data | [21] |
| Management policy | Program selection | Literature research and interviews | Evaluated on performance, with a lag and dependent on the dataset. | [22] |
| | Exploring the role of equity in emergency management | Literature review | Lack of practical considerations | [23] |
| Risk assessment | Development of a disaster evaluation model | Fuzzy evaluation model | Depending on the size of the dataset | [17] |
| | Informing flood hazard assessment | Geographic information decision-making system | Depending on the size of the dataset | [24] |
| | Risk public opinion control | Multi-criteria decision-making | Depending on the size of the dataset | [25] |

construction management should be combined with engineering knowledge. Therefore, the following analysis focuses on emergency construction management from an engineering perspective. Emergency construction management encompasses the effective organization, coordination, supervision, and guidance of personnel, equipment, materials, and the environment at a construction site during emergencies or crises [26]. It aims to ensure construction projects' safety, quality, and progress while minimizing or preventing disasters and accidents. Emergency construction management is a vital component of overall emergency management and carries a substantial responsibility and capability for construction enterprises [27].

Emergency planning is crucial in construction management. An emergency construction plan entails pre-determined response measures and procedures for potential or actual emergencies [28]. Table 2 presents the main research perspectives on the construction and management of emergency hospitals and summarises the contributions and limitations of the research. It is essential to regularly update and improve the emergency construction plan, adjusting and implementing it according to the actual situation. Some scholars have used BIM technology to document and evaluate the emergency response process and results, analyze the efficiency and effectiveness of emergency response, and summarize lessons learned [29]. Furthermore, mathematical models, algorithms, software, and other tools can be employed to solve optimal resource allocation schemes based on resource demand and supply, which ensures the maximization of resource utilization, efficiency, satisfaction, and other relevant indicators [30]. The configuration and scheduling of emergency construction resources should be based on the emergency construction plan and site conditions. It is essential to allocate and utilize various resources reasonably to achieve optimal results [31]. Although these methods provide references for resource allocation during the construction process, they can still not solve the collaboration issues in construction engineering. At the same time, such studies also lack consideration of the external environment. Construction monitoring plays a crucial role in ensuring the safe execution of emergency construction projects [32]. It involves real-time tracking, feedback, analysis, and evaluation of the implementation process, aiming to identify and address issues promptly, enhance the quality and efficiency of the construction, and derive valuable insights and recommendations for improvement. Despite the importance of supervision in emergency construction management, research is scarce in this area. A pressing issue is developing a set of emergency construction management methods for architectural projects that are simple to implement and not dependent on the dataset size, which requires connecting with societal changes, identifying safety incidents, and ensuring the rapid and efficient construction of emergency hospitals.

Table 2. Mainstream research perspectives on emergency construction management.

| Reference | Perspective | Methods | Deficiencies |
|---|---|---|---|
| [27] | Building technology analysis | Case study | Lack of consideration of accident prevention in construction. |
| [12] | Construction fire management | Case study and image recognition | Recognition accuracy depends on the number of samples while being limited to fire. |
| [33] | Social network | Social network analysis (SNA) | Lack of engineering information considerations |
| [34] | Medical wastewater treatment | Expert interviews | Other hazards in emergency hospitals were not considered, and the study methodology was competent. |
| [35] | Emergency construction cost management | Case study | The analysis from a cost perspective favours resource utilization but gives fewer hints about construction safety. |
| [36] | Construction safety | Virtual reality | Failure to consider linkages with society |

## 2.2 Application of SWOT and STPA methods

SWOT is a strategic analysis tool used to evaluate an organization's strengths, weaknesses, opportunities, and threats to develop appropriate strategies [37]. It helps organizations comprehensively understand their internal conditions and external environment, enabling them to identify their strengths and weaknesses and external opportunities and threats. By formulating different strategies based on different situations, SWOT enhances the rationality and scientificity of organizations' strategies, enabling them to achieve their goals and visions. Additionally, SWOT assists organizations in analyzing their market positions, identifying opportunities and threats, and developing effective marketing strategies and plans [38]. Scholars have utilized SWOT to analyze the development strategies of educational institutions and propose a series of reform strategies [39]. Unlike the analysis in Section 2.1, SWOT does not rely on specific data or experiential models. SWOT can provide a quick analysis of the rescue situation in emergencies where time is limited, and the workload is heavy for emergency rescue personnel [40].

STPA is a system safety analysis method used to identify and eliminate factors that may lead to accidents or hazards, thereby improving the system's safety performance. The STPA method can construct an effective control feedback system. STPA analyses complex systems' structure, function, and behaviour in the engineering industry, identifies safety hazards and risk points, and formulates safety constraints and requirements [41]. STPA can help transportation departments or enterprises analyze transportation systems' control structure and process, discover the causes and consequences of transportation accidents, and develop safety control measures and plans [42]. In addition, STPA also has extensive applications in the medical field [43]. Some scholars use STPA to analyze the service processes and equipment of medical institutions or personnel, identify the possibility and impact of medical errors or mistakes, and develop methods for preventing and handling medical errors [44].

Due to the urgent need to construct emergency hospitals, concerns have been raised regarding the reliability of risk and warning mechanisms that rely on experience models. Furthermore, emergency events are relatively rare, resulting in insufficient data for an adequate dataset. Traditional decision models in research do not explicitly address systematic accidents. These studies primarily analyze emergency management from a managerial perspective. However, the construction of emergency hospitals involves completing numerous engineering tasks within a short timeframe, making it challenging for managers to identify every potential issue from a top-down perspective. Therefore, it is essential to clarify the internal and external environment and identify potential hazards that may arise during the construction process of emergency hospitals. Managing from the perspective of potential hazards can help reduce resource wastage and minimize safety accidents.

## 2.3 Emergency hospital construction regulations

Due to the continuous and uncertain nature of the epidemic, corresponding technical standards are needed to guide the deployment and construction of emergency medical facilities to better apply them to epidemic prevention and control. During COVID-19, China issued emergency hospital construction specifications to help control the epidemic. However, different specifications have different requirements for the construction of emergency hospitals. In China, there are few regulations regarding the construction of emergency hospitals, and many industry standards are issued and applied quickly. Therefore, we have summarized the construction specifications for emergency hospitals in China, and the results are shown in Table 3.

Emergency hospitals are often built quickly to respond to outbreaks effectively. Table 3 provides the codes that govern various aspects of emergency hospital construction, such as land

**Table 3. Overview of technical standards for emergency hospitals.**

| Location | Time | Main elements | Shortcomings | References |
|---|---|---|---|---|
| HuBei | February 2020 | (1) Selection of the site location of the existing building for conversion; | (1) Lack of building construction management requirements. | [45] |
| | | (2) Plan layout and quarantine requirements; Structural design and construction; | (2) Lack of emergency management plans | |
| | | (3) Water supply and drainage, electrical system; | | |
| | | (4) Health and safety; | | |
| | | (5) Reference graphic cases of renovation. | | |
| Shangdong | December 2020 | (1) Selection of the site location of the existing building for conversion; | (1) Failure to plan for new emergency hospitals. | [46] |
| | | (2) Plan layout and quarantine requirements; | (2) Only the structural safety of the building was considered, and construction safety was not analyzed. | |
| | | (3) Structural safety and design; | | |
| | | (4) Water supply and drainage, electricity intelligence; | | |
| | | (5) Fire protection, evacuation, garbage collection, and sewage treatment. | | |
| Beijing | March 2022 | (1) Site selection and key points of planning design; | Lack of building construction management requirements. | [47] |
| | | (2) Evaluation and critical points of architectural design; | | |
| | | (3) Structural safety and design; | | |
| | | (4) Water supply and drainage, electricity intelligence, fire protection, and medical technology. | | |
| National | February 2020 | (1) Selection of the site location; | Lack of building construction management requirements. | [48] |
| | | (2) Plan layout and quarantine requirements; | | |
| | | (3) Structural safety and design; | | |
| | | (4) Water supply and drainage, electricity intelligence. | | |

selection, structural safety, electrification, and resource availability. However, there is a lack of regulations and research specifically focused on the engineering management of emergency hospital construction. Therefore, it is crucial to investigate emergency hospitals from an emergency management perspective. This study takes a systematic approach to analyze the construction management of emergency hospitals. Unlike previous research, this study prioritizes compliance with technical specifications as the main objective of construction management. It explores strategies to ensure the successful construction of emergency hospitals within the complex environment of an epidemic.

## 3. The proposed method

### 3.1 Emergency hospital construction claims and conflict analysis

The literature review reveals that the management of emergency hospital construction involves multiple stakeholders, each with their demands and responsibilities. However, previous studies have not thoroughly examined the responsibilities of emergency hospital construction management, making it challenging to implement in practice. Different roles play distinct functions in managing the construction of emergency hospitals, and there are constraints and cooperation among these roles. The key parties involved in the construction process of emergency hospitals include regulatory authorities, construction parties, and the public. Table 4 provides an analysis of the conflicting demands among these critical parties.

According to Table 4, it is evident that the construction process of emergency hospitals involves multiple stakeholders. Conflicting demands from different roles can lead to conflicts, and resolving these conflicts and promoting the progress of emergency hospital construction

**Table 4. Analysis of emergency hospital stakeholder interests.**

| Participants | Interest claim | Conflict of interests |
|---|---|---|
| Governments | Ensure the quality and safety of emergency hospitals to meet medical needs and social expectations. | (1) Government and Construction: Possible quality, progress, and cost conflicts. Between the two parties may result in the construction party adopting strategies of sacrificing quality or safety to improve efficiency or reduce costs, while the supervisory department may adopt strategies of strengthening supervision or punishment to ensure quality or safety. |
| Contractors | It is hoped that an acute care hospital will be built soon to increase revenue and improve its reputation. | |
| Public | They hope to be free from virus infection and have good medical conditions. | (2) Government and Public: There may be conflicts between public interests and personal interests, which may lead to the strategy of the competent authorities to forcibly requisition land, materials, workforce, and other resources to speed up the construction progress. Meanwhile, society may adopt resistance, protest, and litigation strategies to safeguard their rights and interests. |
| | | (3) Construction and public: The conflict between the construction speed and the medical needs of the masses. Due to resource constraints, the construction period of emergency hospitals may be affected, which will conflict with the needs of the masses. |

is an ongoing issue. Due to the need for rapid completion of numerous engineering tasks, the construction of emergency hospitals often requires the collaboration of multiple construction parties. Coordinating the relationships between these parties and allocating resources becomes an urgent problem to be addressed. Simultaneously, managing emergency hospital construction involves the constraints imposed by various levels and fields of management departments. The key to problem-solving lies in coordinating the resources of these departments and ensuring the successful construction of emergency hospitals. Furthermore, the construction of emergency hospitals must consider establishing effective interaction and communication with society to achieve the socialization and humanization of emergency hospital construction.

## 3.2 The proposed method

**3.2.1 The methodological thinking.** Emergency hospital construction safety is a complex project that requires careful attention at every stage of the process to prevent the spread of the epidemic. In previous emergency management research, risk assessment has been commonly used to quantify and analyze risks, often employing operations research techniques. However, the literature review in Section 2 reveals fewer studies on emergency construction safety, while existing studies rely on the dataset size and do not provide a complete construction safety management program. Dynamic analysis and control of accident causes, and hazards are necessary to achieve the goal of rapid construction. Therefore, it is insufficient to analyze emergency hospital construction management solely using traditional emergency management methods.

Emergency hospital construction management aims to rapidly complete emergency hospitals, which are crucial for disaster relief. One of the most critical tasks is identifying key factors that may cause delays and eliminating safety hazards. Due to the schedule requirements of acute hospitals, construction resources need to be deployed in an orderly manner with short notice. Traffic control during disasters can also pose a significant challenge to emergency management [49]. Table 4 shows that the government, society, and construction companies all

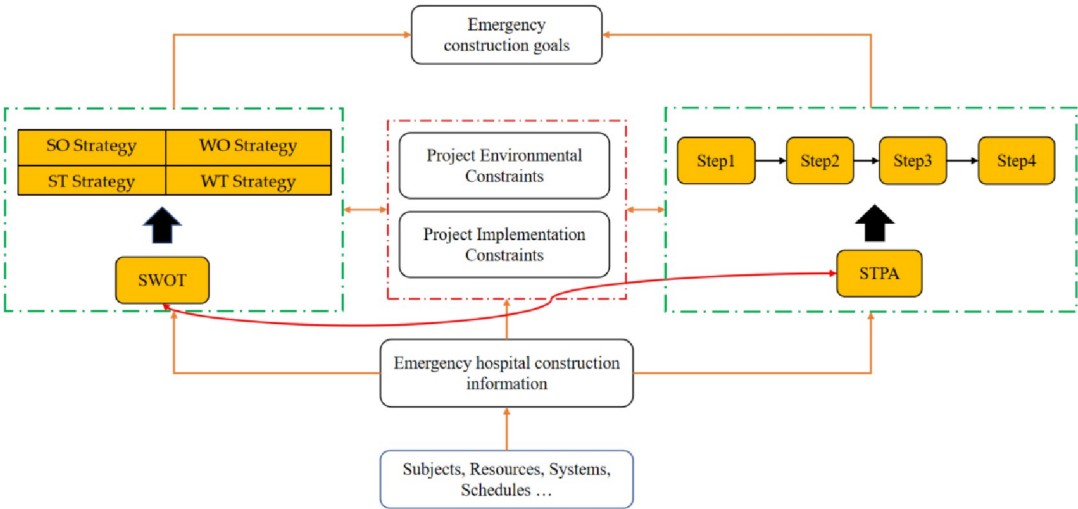

**Fig 1. Emergency hospital construction emergency management structure diagram.**

have demands in constructing emergency hospitals. The current research focuses on coordinating these demands and providing a reasonable strategy for the participation of all parties. As analyzed in Section 2.2, SWOT analysis can assess the external environment and internal conditions without requiring a large amount of data, making it more applicable to emergency management than previous studies. Emergency hospital construction involves numerous engineering tasks that must be completed rapidly, distinguishing it from traditional construction management. Section 2.1, identified through the literature review, reveals that traditional research primarily focuses on construction management from managers' perspective. This approach necessitates managers to gather extensive data and invest a significant amount of time, which does not align with emergency management requirements. In this paper, we employ STPA to analyze emergency construction management from a potential hazard standpoint, offering a more suitable approach for emergency construction management. The specific method is illustrated in Fig 1.

**3.2.2 Steps of SWOT and STPA.** SWOT analysis is a scientific method to analyze the research subject's strengths, weaknesses, opportunities, and threats, which can help the research subject develop a suitable strategic plan [50]. The steps of SWOT analysis are as follows:

Step 1: Confirm the current strategic goals and objectives and clarify the object and scope of the analysis. Collect and organize internal and external information related to the object of analysis. Based on this information, identify internal strengths (S), weaknesses (W), external opportunities (O), and threats (T).

Step 2: Arrange the four elements of S, W, O, and T in a matrix form to create a SWOT analysis table (Table 5). Utilize this table to generate feasible strategic solutions with four main strategies: SO strategy: Seize the opportunity to maximize development by leveraging the advantages. WO strategy: Improve or overcome the disadvantages and strive for opportunities to achieve transformational development. ST strategy: Utilize advantages to cope with or avoid threats and achieve sound development. WT strategy: Reduce or eliminate disadvantages, avoid or minimize threats, and achieve survival and development. Finally, develop a

**Table 5. SWOT strategy table.**

| SWOT | Strengths (S) | Weaknesses (W) |
|---|---|---|
| Opportunities (O) | SO Strategy | WO Strategy |
| Threats (T) | ST Strategy | WT Strategy |

specific action plan based on the strategic plan. Monitor, dynamically adjust, and improve the program according to the action plan.

STPA takes a systems perspective on studying security. One notable feature is that it does not rely on empirical models like management science methods like the Delphi method. Furthermore, STPA allows for analyzing interactions between systems. The STPA method typically involves four steps [51, 52]. The process of the STPA method is shown in Fig 2.

Step 1: Defining system boundaries, objectives, and constraints is the initial step in determining the scope, functionality, performance, interfaces, environment, and security goals and constraints the system must adhere to. This step involves: (1) Identifying losses. (2) Identifying hazards at the system level. (3) Recognizing common errors associated with system-level hazards. (4) Establishing system-level constraints.

Step 2: Constructing the control structure is crucial in ensuring effective control over the system's actions. The primary aim of this step is to develop a hierarchical control model that incorporates feedback loops. Fig 2 illustrates the hierarchical control structure. The control algorithm represents the decision-making process, while the process model reflects the previous decisions' internal concepts and can be updated based on the state of the controlled process.

Step 3: Identifying unsafe control actions is crucial in ensuring safety in specific situations and worst-case scenarios. Hazards can range from personal injury or life-threatening issues to

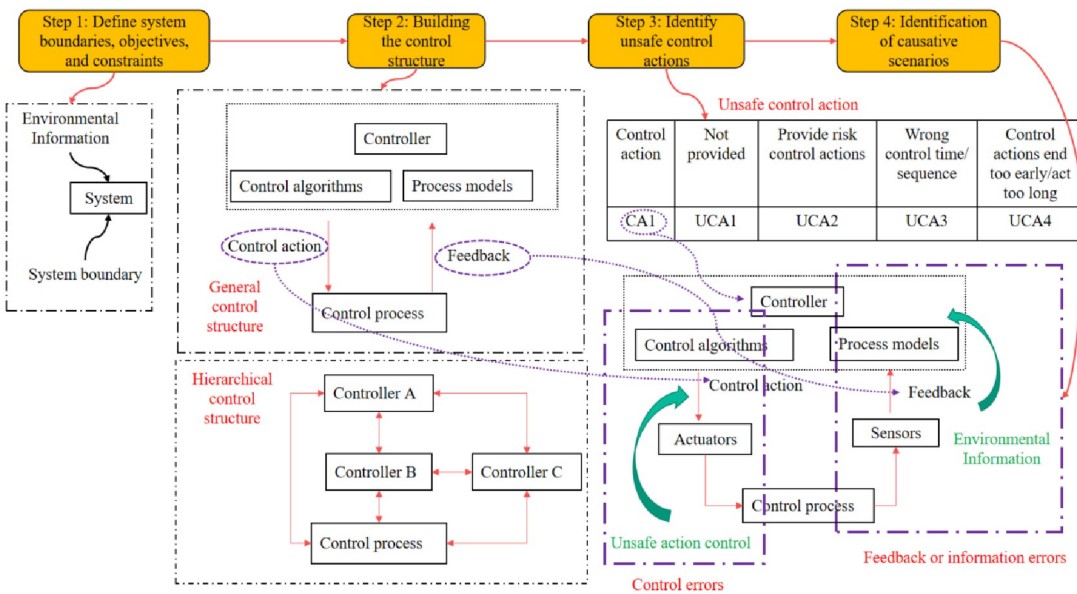

**Fig 2. STPA method flow chart.**

mission failure, performance failure, and environmental losses. Describing the context in which a control action is considered unsafe is essential. Consistently unsafe control actions are not incorporated into the system design. Each unsafe control action should specify the circumstances or scenarios in which it poses a risk. Four conditions indicate potential risk for the system: (1) Failure to provide a control action results in risk. (2) Providing the control action leads to a hazard. (3) Potentially safe control actions are provided, but the timing or execution is incorrect. (4) The duration of the control action is either too long or too short (only applicable for continuous control, not discrete).

Step 4: Identification of causative scenarios: Causal scenarios describe the triggers that may lead to unsafe control actions and hazards. Based on this, the causal scenario must be considered from two perspectives: (1) Why does the unsafe control action occur? (2) Why does the control action appear poorly executed and lead to a hazard?

## 4. Case study

The validity and operationalization of the methodology can be illustrated by selecting appropriate cases. The construction of emergency hospitals is an essential option in disaster emergency management. A case study is selected for Wuhan Vulcan Hill Hospital, a typical example of a hospital built quickly and with much construction work.

### 4.1 Case introduction

Wuhan Vulcan Hill Hospital is located in Zhiyin Lake Avenue, Haidian District, Wuhan City. It is a unique hospital established within the Wuhan Staff Sanatorium, following the model of the 'Beijing Xiaotangshan Hospital used during the SARS outbreak 2003. The hospital's primary focus is on treating patients with Covid-19. Table 6 shows the specific details of the Vulcan Hill Hospital [53].

### 4.2 SWOT analysis of Vulcan Mountain Hospital construction process

According to Table 6, there was collaborative involvement from various construction parties during the construction process of Vulcan Mountain Hospital. The government departments provided much assistance in the construction of Vulcan Mountain Hospital, and at the same time, members of society actively participated in the construction. However, Vulcan Mountain Hospital still faces issues such as construction coordination and limited resources. The results of the SWOT analysis are as follows:

Internal advantages: The third Bureau of CSCEC (China State Construction Engineering Corporation), a general construction contractor, possesses extensive construction experience. In addition, they employ advanced technologies such as BIM+, 5G, and intelligent construction to achieve high-level construction practices, including design and construction integration, modular productization, digitalization, and intelligent management.

Internal Disadvantages: The construction of Vulcan Mountain Hospital took place during the Spring Festival and faced shortages and challenges in resources, including workforce, supplies, and equipment, which required transportation assistance from all over the country. The construction was carried out under difficult circumstances, with a complex construction site environment, a high-density construction crew, and a tight construction schedule, posing multiple safety hazards and accident risks. Regarding quality control, the accelerated completion of construction work has increased demands on construction personnel and management.

**Table 6. Vulcan Mountain Hospital construction information sorting.**

| Construction Stage | Time | Participants | Problems faced and goals accomplished |
|---|---|---|---|
| Received a construction assignment | January 23, 2022 | (1) Wuhan urban and rural construction bureau | China Zhongyuan provided drawings of Xiaotangshan Hospital. |
| | | (2) China construction third bureau | |
| Start of site formation | January 23, 2020, at 22:00 | (1) China construction third bureau (2) Wuhan construction industry, (3) Wuhan municipal government, (4) Hanyang municipal government. | (1) The construction site was on a mud pond and hill, with a foundation height differential of about 10 meters, making the construction environment highly undesirable. |
| | | | (2) The construction team needed to level off seven soccer fields of open area in such a location. Common sense estimations The quantity of work to take at least one or two months. Moreover, there are high-voltage lines above the site and gas and water pipelines on the ground, so relocation is also tricky. |
| Establishment of on-site command | January 24, 2020, at 1:00 am | (1) China Construction Third Bureau | At the beginning of the construction phase, during the Chinese New Year, there were tremendous challenges in transportation, material distribution, and staff rationing. |
| Assemble construction personnel and materials. | January 24 to January 30, 2020 | (1) China Construction Third Bureau | Two thousand people worked concurrently at the peak construction site, including hundreds of design crafts. |
| | | (2) Social assistance | |
| Completion of prominent structure erection | January 29 to February 1, 2020 | (1) China Construction Third Bureau | The hospital adopted the construction design of "two cloths and one membrane" + wastewater closure treatment: cloth, i.e., geotextile, is often a permeable material that protects the soil from erosion; HDPE membrane (high-density polyethene film) is a waterproof material used for waterproofing and seepage control of the foundation and in the top and bottom of the impermeable layer also laid 20cm sand each, with a robust anti-seepage effect. Hospitals employ the modulus of 33, the highest modularity, with 36 panels made to build. Three container boards are united to make two wards, and the container boards in the hallway are vertical to the wards. |
| | | (2) China Railway Eleventh Bureau | |
| | | (3) China Metallurgical Group. | |
| Complete the installation of electric power, communication, and other supporting facilities. | January 31 to February 2, 2020 | (1) Wuhan Caidian district power supply company | (1) Vulcan Hill Hospital must ensure the normal functioning of the power supply, water supply, gas supply, communication, and other operations. |
| | | (2) Telecommunications companies | (2) Wuhan caidian district power supply Company is responsible for power engineering construction. |
| | | | (3) Telecommunications companies are responsible for constructing and optimizing the communication network to ensure the hospital's seamless and unobstructed internal and external communication. |
| Official delivery | February 2, 2020 | (1) Wuhan Municipal Government | They were constructed and delivered with a total construction time of 10 days. |
| | | (2) State-owned Assets Supervision | |
| | | (3) Administration commission of the state council | |

External opportunities: The government supported the construction of Vulcanshan Hospital as a significant project to address national public health emergencies. Relevant policies enhanced this support, making the hospital more accessible and favourable. Additionally, the staff at XiaoTangShan Hospital shared the construction plans and provided technical assistance. Furthermore, the community has displayed a strong interest in and support for the hospital's development.

External threat: Vulcan Hill Hospital was established to respond to the epidemic and underwent rapid changes during the initial construction phase. The construction of the emergency hospital required careful consideration of the evolving dynamics of the epidemic, leading to swift modifications in construction plans and actions to align with the demands of epidemic prevention and control. Additionally, being situated at the border of the lake, Vulcan

Table 7. Vulcan Hill Hospital SWOT strategy analysis.

| | Strengths (S) | Weaknesses (W) |
|---|---|---|
| Opportunities (O) | **SO Strategy**: Utilize policy support and social aid to promote quality assurance in emergency hospitals. Enhance construction efficiency and quality through the accumulation of experience. | **WO Strategy**: Utilize government support and social aid to mitigate the constraints of limited resources and security concerns. Enhance quality control measures and monitoring protocols to ensure effective implementation. |
| Threats (T) | **ST Strategy**: In the face of challenging diseases and environmental consequences, it is crucial to overcome stress and leverage it as an opportunity to achieve the goal of constructing emergency hospitals. | **WT Strategy**: Avoid delays and mishaps owing to resource restrictions and safety hazards. Prevent poor outbreak control owing to insufficient quality control. |

Mountain Hospital had to address impermeability, moisture-proofing, and ecological safety concerns. Moreover, as an emergency hospital in the early stages of the epidemic, the successful construction of Vulcan Hill Hospital played a crucial role in boosting confidence in the ongoing battle against the virus.

The analysis can be about the emergency management SWOT strategy of Vulcan Mountain Emergency Hospital. Contingency strategies are shown in Table 7. Following the actual situation, the construction manager of Vulcan Mountain Hospital should actively coordinate resources and relevant stakeholders and select appropriate response strategies.

### 4.3 STPA analysis of Vulcan Mountain Hospital construction process

The Vulcan Mountain Hospital is a typical prefabricated building with a large construction volume and a short construction period. Modular design can provide diversified treatment plans and different medical conditions for the hospital. The construction of Vulcan Mountain Hospital will be analyzed from the perspective of prefabricated buildings using STPA analysis. According to the location of the hospital and the technical specifications of the construction technology management, the technical specifications of Hubei Province are mainly referenced [45].

**4.3.1. Identify system-level incidents and hazards.** The level of safety system loss in the construction was determined based on the strict time requirements for emergency work and the necessity to lift and stack the assembled building. Additionally, the needs of the emergency hospital, such as the requirement for electrification and unobstructed emergency access, were considered. The SWOT analysis results provide strategic references for STPA and serve as the basis for construction time analysis. The results are presented in Table 8.

System-level hazard refers to the state of the system being in the most unfavourable environment, which may lead to a system-level accident. Once the components are manufactured in the factory, they need to be transported to the construction site for stacking and subsequent lifting during the construction process. Considering the characteristics mentioned above of assembled building construction, a categorization of system-level hazards was conducted, and the results are shown in Table 9.

**4.3.2 Building the control structure.** The control structure plays a crucial role in the STPA method for constraining hazards. It comprises four components: the controller, actuator, control process, and sensing feedback. The controller issues commands to the actuators, while the sensing feedback monitors the actuator's course of action and relays information back to the controller.

Based on Table 6 and the SWOT analysis, it can be inferred that in the construction process, in addition to meeting the characteristics of prefabricated construction, the issue of collaboration among all stakeholders needs to be considered. To successfully construct the Wuhan Vulcan Hill Emergency Hospital, a general contracting building, the contracting company

**Table 8. System level losses.**

| System loss number | System losses | References |
|---|---|---|
| L-1 | Dropping objects from a height | [54, 55] |
| L-2 | Object collision strike | [55, 56] |
| L-3 | Mechanical injury | [54, 57] |
| L-4 | Personnel casualty | [58, 59] |
| L-5 | Time delay | [60, 61] |
| L-6 | Cannot meet emergency hospital requirements | [6, 62] |

collaborated with design units, suppliers, and subcontractors. This collaboration allowed for the rapid completion of the construction through a reverse design approach (design as you build). In line with the unique construction model of the comprehensive emergency hospital, we analyzed the control structure of Vulcan Mountain Hospital using the STPA method. The control structure diagram is shown in Fig 3.

**4.3.3 Identify unsafe control actions.** The Vulcan Emergency Hospital uses a unique "design and build" approach. This study focuses on the characteristics of assembly building construction and Vulcan Mountain Hospital and categorizes the unsafe control actions into four aspects: technical handover, quality acceptance, mechanical assembly inspection and deployment, and component lifting and installation adjustment. A detailed analysis was conducted following the implementation steps of the STPA method to obtain the results of unsafe control actions during the construction process of an emergency hospital. The specific results are shown in Table 10.

**4.3.4 Identification of causative scenarios.** In section 4.2, the possible unsafe control actions during construction are categorized and sorted out in conjunction with the emergency hospital building structure. The causes of the unsafe control actions are next sorted according to the steps of the STPA method. According to Table 3, many parties are involved in the construction process of emergency hospitals, and they are responsible for different tasks, which will involve issues of coordination between different units. According to SWOT analysis, although the public strongly supports the construction of emergency hospitals, potential threats and internal unfavourable conditions will still impact the construction process. The analysis is as follows:

1. Technical presentation. The technical briefing ensures that every individual involved in construction activities understands the project's specific conditions, organization, technical requirements, and critical measures. It allows for a systematic understanding of the construction process and its critical positions. Many staff working simultaneously without technical briefing may result in irreparable losses. By providing a detailed project overview, the technical briefing enables workers to comprehend their specific tasks, operation

**Table 9. System-level hazards and their corresponding system losses.**

| System-level hazard number | Hazard description | Losses caused |
|---|---|---|
| H-1 | Tipping of prefabricated elements | L-1, L-2, L-4, L-5 |
| H-2 | Failed lifting of prefabricated components | L-1, L-2, L-3, L-4, L-5 |
| H-3 | Poorly connected prefabricated elements | L-5, L-6 |
| H-4 | External bracket not secure | L-1, L-2, L-4, L-5, L-6 |
| H-5 | Management deficiencies | L-1, L-2, L-3, L-4, L-5, L-6 |

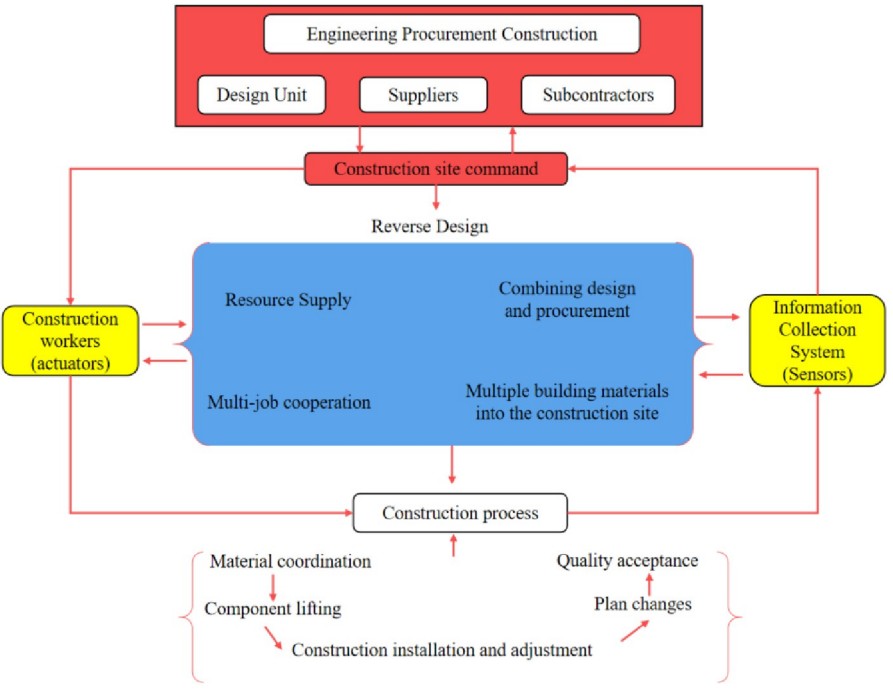

**Fig 3. Vulcan Mountain Emergency Hospital control structure.**

methods, construction techniques, quality standards, and safety precautions. This clarity empowers construction operators to perform their tasks efficiently and promotes orderly construction, ultimately reducing quality defects and enhancing construction quality.

2. Quality acceptability. Quality acceptance is a crucial management tool to ensure the safety of a project and achieve the objective of safe production. It is essential to remove safety dangers promptly and prevent safety incidents. Routine quality acceptance is necessary, and the inspection team should include representatives from all levels of the construction organization. Without comprehensive quality inspection, the project's quality cannot be guaranteed.

**Table 10. Emergency hospital construction unsafe control action.**

| Control actions | Failure to provide control actions lead to danger | Provide risk control actions | Wrong control time/sequence | Control actions end too early/act too long |
|---|---|---|---|---|
| Technical presentation | *UCA1*: No technical briefing before construction (H-1,2,3,4,5) | *UCA2*: Errors or lack of care in technical delivery (H-1,2,3,4,5) | N/A | N/A |
| Quality acceptance | *UCA3*: No quality acceptance (H-3,4,5) | *UCA4*:Performing quality inspection without careful acceptance (H-3,4,5) | N/A | N/A |
| Mechanical assembly inspection and deployment | *UCA5*:No mechanical inspection No mechanical Deployment (H-2,5) | *UCA6*:Incomplete mechanical inspection (H-1,2,5)<br>*UCA7*:Perform incomplete mechanical deployments (H-2,5) | *UCA8*:Failure to inspect and deploy machinery before starting work (H-1,2,5) | *UCA9*:Failure to redeploy machinery in accordance with planned changes (H-1,2,5) |
| Component lifting and installation adjustment | *UCA10*:No lifting Signal provided (H-2,3,5) | *UCA11*:Providing the wrong lifting signal (H-2,3,4,5) | *UCA12*:Lifting signals occur in the wrong order (H-2,3,5) | *UCA13*:Lifting is not completed, lifting signal has been extinguished (H-2,5) |

3. Mechanical assembly inspection and deployment. Construction equipment plays a vital role in emergency engineering construction procedures. Firstly, relying solely on the human workforce to enter colossal emergency sites is risky. Using construction equipment as auxiliary tools can ensure the safety of both rescuers and the rescued. One crucial aspect of combating the rapid spread of epidemics is the early construction of emergency hospitals. As an assembly-type prefabricated hospital, emergency hospitals require numerous prefabricated components to be coordinated with machinery. Therefore, it is essential to deploy the equipment appropriately and regularly inspect its status to ensure construction safety and timely project completion. Any equipment breakdown, lifting accidents, casualties, or component damage can delay the emergency hospital's construction schedule and negatively impact epidemic management.

4. Component lifting and installation adjustment. The proper sequencing and adjustment of components is crucial for the progress of constructing the assembly building. Ending the lifting guidelines too early or too quickly can lead to collisions and damage to components, resulting in losses and even casualties.

## 5. Discussion

This study aims to address the limitations of previous research by focusing on the construction management of emergency hospitals from the perspective of potential risks. Unlike traditional studies that primarily consider the viewpoint of managers, this research aims to meet the time requirements of emergency hospitals better and effectively analyze and manage various potential hazards. It takes into consideration the source of danger and proposes targeted control structures. We discuss the methods' applicability and the insights the research brought, hoping to provide references for more emergency management studies.

### 5.1 Applicability of methods

SWOT analysis is a helpful tool for conducting problem analysis from a macro level, as it helps to summarize numerous influencing factors. However, it has certain limitations in addressing qualitative deficiencies. On the other hand, STPA focuses on dynamic relationships within a system and provides a more detailed analysis. By combining both methods, problem evaluation can be approached from different perspectives, enhancing analytical integrity.

For emergency hospital construction managers, SWOT analysis can be utilized to identify strengths such as rapid response, flexible adjustment, and professional teams. It also helps identify weaknesses like inadequate equipment, staff shortages, and lack of standardized procedures. Additionally, SWOT analysis can reveal opportunities in terms of policy support, social attention, technological innovation, and threats posed by changes in epidemic situations, natural disasters, and social unrest. By conducting a SWOT analysis, emergency hospital construction managers can formulate strategic plans and specific measures that align with their strengths, overcome weaknesses, leverage external opportunities, and mitigate external threats. This approach allows them to adapt to their specific situations and external trends.

The construction of emergency hospitals requires collaboration among various personnel. STPA, which focuses on dynamic relationships within a system, addresses the limitations of SWOT analysis in detail. STPA can assist emergency hospital construction managers in determining system boundaries, control structures, behaviours, and constraints. It can also help identify control deficiencies that may result in deviations from or failures of objectives. Additionally, STPA can identify potential risks and hazards within the system and suggest

preventive and improvement measures. For example, through the STPA analysis, managers can understand the composition of the departments involved in constructing management systems. It also includes command centres, on-site command posts, and various working groups as control structures. Control behaviours involve planning, design, construction, operation, and maintenance, while constraints involve time limitations, quality requirements, and safety standards. Subsequently, emergency hospital construction managers can identify control deficiencies, such as unclear instructions, untimely information, and lack of coordination, and propose corresponding preventive and improvement measures. These measures may include clarifying responsibilities, enhancing communication and collaboration, and conducting training and drills.

SWOT analysis and STPA analysis complement each other, providing several advantages. By combining both methods, problem evaluation can be approached from different perspectives, enhancing analytical integrity. SWOT analysis and STPA analysis are based on systems thinking, acknowledging the complexity of emergency hospital construction management. They emphasize the need for analysis from multiple perspectives, including the whole and the parts, the static and the dynamic, and the internal and the external. The combination of SWOT analysis and STPA analysis allows emergency hospital construction managers to gain a macro-level understanding of the system's overall situation and development trends while also analyzing the details and dynamic relationships at a micro level. This comprehensive approach enhances the completeness and comprehensiveness of the analysis. Additionally, the combination of SWOT analysis and STPA analysis enables emergency hospital construction managers to qualitatively identify the system's strengths, weaknesses, opportunities, and threats and quantitatively evaluate the risks and hazards. Furthermore, the combination of SWOT analysis and STPA analysis facilitates the formulation of system objectives and plans from a strategic perspective and enables the implementation of system control and improvement from an operational perspective.

## 5.2 Management insights

Insights on management can be derived from the analysis of emergency hospital construction. Fig 4 presents the management relationship between disaster information, resources, and collaboration. This relationship can be examined from three perspectives: planning background, process, and practice. In terms of the planning background of emergency rescue force construction, it is crucial to ensure overall development, security, and the integration of various high-quality resources. Therefore, emergency rescue needs to define the goals, tasks, standards, and guarantee mechanisms of emergency construction, aiming for scientific, standardized, and professional development of emergency rescue force construction. Regarding the process of emergency hospital construction, it is evident that rapid response, overcoming challenges, and achieving quick results are essential. Thus, project management should be strengthened, workflow optimized, and efficiency and quality improved to attain efficient, refined, and standardized development of emergency hospital construction. As for the practice of emergency hospital operations, it is necessary to coordinate the mobilization of medical and health institutions, establish regional linkages, and manage personnel under emergency conditions. Additionally, establishing a sound hierarchical, layered, and diversionary mechanism for treating major epidemics is crucial. Coordinating resource allocation, enhancing information sharing, and improving collaboration capabilities will contribute to coordinating, intelligence, and humanising emergency hospital operations, thereby facilitating effective management of emergency hospital construction.

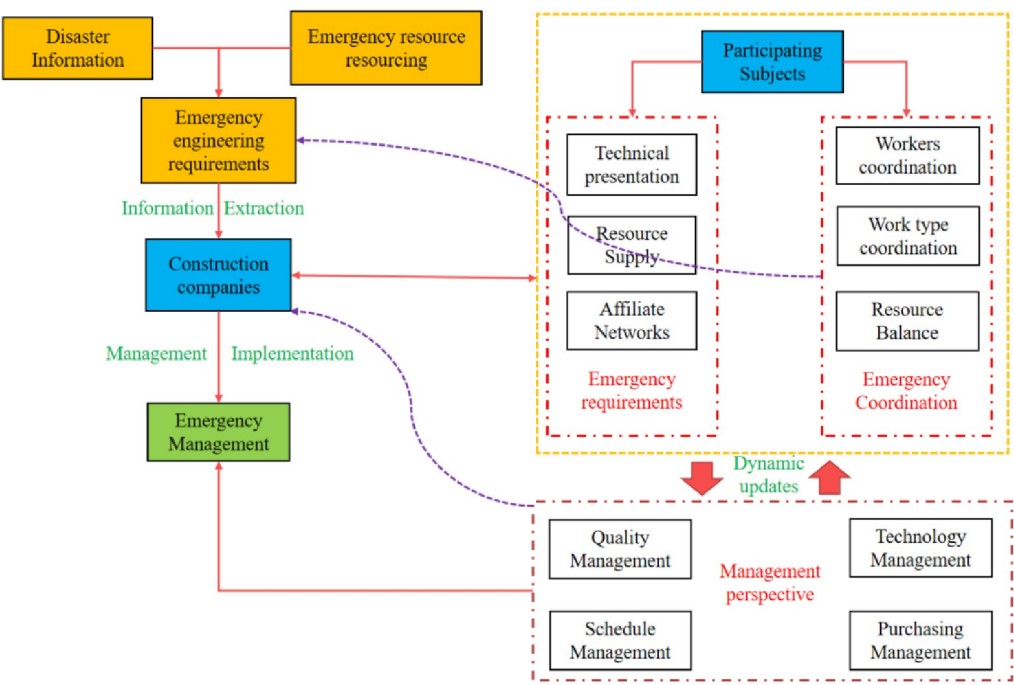

**Fig 4. Emergency hospital emergency management ideas.**

## 5.3 Limitations

This article introduces the SWOT-STPA method as a reference for emergency hospital construction management, which differs from previous studies. However, it also acknowledges some limitations. Firstly, the SWOT analysis relies on information gathering and may be influenced by subjective factors, even though it does not require extensive data for judgment. Secondly, while STPA focuses more on engineering and technical issues, the construction management of emergency hospitals may also involve non-technical factors. Therefore, it is crucial to consider non-technical factors and maintain objectivity in future research. For instance, text mining can be utilized to collect information, minimise subjective errors and consider the impact of personnel qualities and cultural levels on the construction of emergency hospitals.

## 6. Conclusion

Emergency hospital construction management is different from traditional building management methods. If we approach it from the manager's perspective, it will take much time and not meet the timeliness of emergency management. This study focuses on the perspective of the accident source and uses the SWOT-STPA method to study emergency hospital construction management. The example of Vulcan Mountain Hospital demonstrates its effectiveness, which provides a new perspective and new methods for emergency hospital construction management.

## Supporting information

**S1 File. Hospital construction technical requirements reference.**
(DOCX)

## Author Contributions

**Conceptualization:** Dongliang Zhu.

**Data curation:** Dongliang Zhu.

**Formal analysis:** Dongliang Zhu, Song Wang, Yaru Li.

**Methodology:** Dongliang Zhu.

**Project administration:** Song Wang.

**Software:** Dongliang Zhu.

**Visualization:** Dongliang Zhu.

**Writing – original draft:** Dongliang Zhu.

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
