## [Decision Letter · Decision Letter 0]

9 Oct 2023

PONE-D-23-27781Strategic Management and Risk Control of Emergency Hospital Construction: SWOT and STPA framework from a Systems Thinking PerspectivePLOS ONE

Dear Dr. Zhu, 

Thank you for submitting your manuscript to PLOS ONE. After careful consideration, we feel that it has merit but does not fully meet PLOS ONE’s publication criteria as it currently stands. Therefore, we invite you to submit a revised version of the manuscript that addresses the points raised during the review process.

ACADEMIC EDITOR:One of the reviewer's recommended a "rejection" for the article. The contributions of the article seems incremental with very little emphasis on the contributions of the article towards global construction engineering and management discipline and/or systems engineering. Authors should clearly specify the contributions and how this study is unique and contributes to the literature.   The application of methods seems to not be specific for this this particular problem. What is the rationale for choosing them in this problem context (i.e., rapid construction)? 

We look forward to receiving your revised manuscript.

Kind regards,

Sudipta Chowdhury

Academic Editor

PLOS ONE

 [This study was supported by the Xinyang Normal University Graduate Research Innovation Fund (2022KYJJ089)].  

4. We note that Figure 3 and 4 in your submission contain copyrighted images. All PLOS content is published under the Creative Commons Attribution License (CC BY 4.0), which means that the manuscript, images, and Supporting Information files will be freely available online, and any third party is permitted to access, download, copy, distribute, and use these materials in any way, even commercially, with proper attribution. For more information, see our copyright guidelines: http://journals.plos.org/plosone/s/licenses-and-copyright.

A. You may seek permission from the original copyright holder of Figure 3 and 4 to publish the content specifically under the CC BY 4.0 license. 

B. If you are unable to obtain permission from the original copyright holder to publish these figures under the CC BY 4.0 license or if the copyright holder’s requirements are incompatible with the CC BY 4.0 license, please either i) remove the figure or ii) supply a replacement figure that complies with the CC BY 4.0 license. Please check copyright information on all replacement figures and update the figure caption with source information. If applicable, please specify in the figure caption text when a figure is similar but not identical to the original image and is therefore for illustrative purposes only.

Reviewers' comments:

Reviewer's Responses to Questions

**Comments to the Author**

1. Is the manuscript technically sound, and do the data support the conclusions?

Reviewer #1: Yes

Reviewer #2: No

2. Has the statistical analysis been performed appropriately and rigorously? 

Reviewer #1: N/A

Reviewer #2: No

3. Have the authors made all data underlying the findings in their manuscript fully available?

Reviewer #1: Yes

Reviewer #2: Yes

4. Is the manuscript presented in an intelligible fashion and written in standard English?

Reviewer #1: Yes

Reviewer #2: Yes

5. Review Comments to the Author

Reviewer #1: This research study appears to address an important topic, which is the effective management of emergency prefabricated building construction, particularly in the context of emergency hospitals during disasters. However, there are several critical aspects that need to be reviewed and criticized:

1. Overemphasis on Methods: The study heavily emphasizes the use of SWOT and STPA methods but does not adequately explain why these methods were chosen and their relevance and applicability to emergency hospital construction.

2. Limited Generalizability: The study primarily focuses on a single case study, which may limit the generalizability of the findings. It's essential to discuss how the insights gained from the Vulcan Hill Hospital case can be applied to other emergency hospital construction projects or disaster scenarios.

3. Unaddressed Limitations: The study acknowledges some limitations, such as not considering dynamic changes in emergency hospital construction. However, it does not provide recommendations for addressing these limitations. A more robust discussion of limitations and suggestions for future research would enhance the study's quality.

4. Assumption of Inadequacy of Traditional Methods: The paper makes a strong assertion that traditional safety assessment methods are insufficient for emergency hospital construction without providing sufficient evidence to support this claim. It should elaborate on why traditional methods are inadequate and how the proposed method addresses these shortcomings.

In summary, while the research addresses an important topic related to emergency hospital construction management, it requires some improvements to address the above points.

Reviewer #2: In this manuscript, the SWOT and STPA methods were implemented to increase effectiveness and reduce risks in the construction management of emergency buildings. The case study to investigate the proposed method was the Wuhan Vulcan Mountain Hospital construction in China during the COVID-19 outbreak.

The manuscript was written well, and the flow of information was smooth.

The hospital was constructed in 10 days. Many aspects, such as governmental supports and land conditions, were discussed, but an in-depth analysis is missing. The proposed method failed to identify and offer solutions for the hospital's construction shortcomings. If the technique applied to construct the hospital is effective, what is the necessity of the proposed method?

Besides, the SWOT and STPA are well-established methods.

Emergency hospital emergency management idea was proposed in Figure 6. A detailed description of the proposed idea was not given. Besides, how the proposed idea is unique compared to the method that was used to construct Wuhan Vulcan Mountain Hospital is not given.

Overall, the contribution of the study is minor.

6. PLOS authors have the option to publish the peer review history of their article (what does this mean?). If published, this will include your full peer review and any attached files.

Reviewer #1: No

Reviewer #2: No

---

## [Author Response · Author response to Decision Letter 0]

10 Nov 2023

Review 1

Dear Reviewer:

Thank you for your excellent comments, we have carefully considered each of them and revised the corresponding sections, and the specific responses have been itemized. For your reviewing convenience, we have redlined the changes, with major revisions labeling the chapter headings and minor revisions labeling the content.

Comment 1:

Overemphasis on Methods: The study heavily emphasizes the use of SWOT and STPA methods but does not adequately explain why these methods were chosen and their relevance and applicability to emergency hospital construction.

Response 1: 

We thank the reviewer for this good suggestion. We apologize for the lack of explanation regarding the method selection and analysis of method applicability in the original text. Based on your suggestions and our understanding, we have made the following modifications to the article: (1) We added Section 2.3, which describes the main applications of SWOT and STPA. (2) Section 3.1 was added to analyze the benefits of the emergency hospital construction process. In section 3, the main issues and applications of the emergency hospital construction process are analyzed. (3) Section 3 of the original text was transformed into Section 3.2 and the relevance and rationality of the methodology was analyzed in 3.2.1.

Comment 2:

Limited Generalizability: The study primarily focuses on a single case study, which may limit the generalizability of the findings. It's essential to discuss how the insights gained from the Vulcan Hill Hospital case can be applied to other emergency hospital construction projects or disaster scenarios.

Response 2: 

We thank the reviewers for this excellent suggestion. The manuscript indeed lacks relevant discussions. We have reviewed the case study and added Section 5, where we conducted an analysis on the application of methods, managerial insights, and research limitations. Once again, we would like to express our gratitude for the valuable feedback from the reviewers.

Comment 3:

Unaddressed Limitations: The study acknowledges some limitations, such as not considering dynamic changes in emergency hospital construction. However, it does not provide recommendations for addressing these limitations. A more robust discussion of limitations and suggestions for future research would enhance the study's quality.

Response 3: 

We thank the reviewer for this good suggestion. We address the shortcomings of SWOT and STPA in Section 5.3 by presenting the issues that need to be addressed in future research and presenting ideas to address future issues.

Comment 4:

Assumption of Inadequacy of Traditional Methods: The paper makes a strong assertion that traditional safety assessment methods are insufficient for emergency hospital construction without providing sufficient evidence to support this claim. It should elaborate on why traditional methods are inadequate and how the proposed method addresses these shortcomings.

Response 4: 

We thank the reviewer for this good suggestion. In our literature review section in the second part, we have conducted an analysis on the main research perspectives of emergency management and emergency construction management. We have summarized the findings through Table 1 and Table 2. In the literature review, we have also summarized the main uses of SWOT and STPA. Additionally, in Sections 3.1 and 3.2.1, we have analyzed the applicability and relevance of the methods.

Summary of responses to reviewer 1

I would like to express sincere gratitude on behalf of all the authors for your comments. We have carefully considered each of your comments and made reasonable modifications to address the issues you raised. We hope you will allow us to publish this article on PLOS. Of course, if you have any further questions, feel free to ask us at any time.

Review 2

Dear Reviewer:

Thank you for your valuable comments. We apologize for causing you confusion in regards to this article. We understand that your comments were not optimistic and lacked specific details, however, the editor has given us an opportunity to make revisions, so we would like to attempt to do so. We have addressed your comments point by point and explained the changes we have made that you did not mention. We hope you can give us another chance. Thanks again for all your hard work. For your reviewing convenience, we have redlined the changes, with major revisions labeling the chapter headings and minor revisions labeling the content.

Comment 1:

The hospital was constructed in 10 days. Many aspects, such as governmental supports and land conditions, were discussed, but an in-depth analysis is missing.

Response 1：

We thank the reviewer for this good suggestion. Firstly, we have conducted a more detailed analysis of the case study process in the manuscript, adding more detailed content and referring to the construction standards for emergency hospitals. Secondly, we have added an analysis of the problems faced during the construction process of emergency hospitals in section 3.1. In section 3.2.1, we have discussed the applicability of the methods. In our revision, an analysis of the applicability of the methodology was added to increase the persuasiveness.

Comment 2:

If the technique applied to construct the hospital is effective, what is the necessity of the proposed method? Besides, the SWOT and STPA are well-established methods.

Response 2：

We thank the reviewers for this excellent suggestion. In sections 2.1 and 3.1, we conducted relevant literature review and analyzed the issues faced by emergency construction management. Through the analysis in sections 2.1 and 3.1, it was found that the current research lacks analysis from the perspective of hazard sources. We utilized a combined approach of SWOT and STPA for analysis. We acknowledge your viewpoint that SWOT and STPA are indeed stable and effective methods, but we believe that SWOT and STPA have a complementary relationship. In sections 3 and 5, we provide a detailed analysis of the necessity.

Comment 3:

Emergency hospital emergency management idea was proposed in Figure 6. A detailed description of the proposed idea was not given.

Response 3：

We thank the reviewer for this good suggestion. We apologize for the lack of explanation about Figure 6. Due to the deletion of two copyrighted images, Figure 6 is now changed to Figure 4. In Section 5.2, we combined Figure 4 with our research process to provide management recommendations for emergency construction management from the perspective of preventing hazardous sources. We hope that this can serve as a reference for future exploration. Section 5 also includes a possible analysis of method application and research limitations, which can provide references for similar studies and offer additional insights.

Comment 4:

How the proposed idea is unique compared to the method that was used to construct Wuhan Vulcan Mountain Hospital is not given. Overall, the contribution of the study is minor.

Response 4：

The reviewer suggests a very interesting topic. We thank you for your excellent review and we would be happy to discuss this topic with you.

Firstly, you mentioned the difference between the method we used and the construction of Wuhan Vulcan Mountain Hospital. In section 2, we analyzed the building codes. Our research is based on the building codes and uses STPA for the analysis from the perspective of hazard sources. On the other hand, the construction of Wuhan Vulcan Mountain Hospital was based on the experience of the construction unit and formed from the perspective of the manager, as there were no relevant technical specifications at that time. The two differ in terms of research perspective. The analysis from the perspective of hazard sources is more suitable for the characteristics of emergency buildings. In section 5, we also analyzed relevant issues. 

Secondly, in terms of methodology, Wuhan Vulcan Mountain Hospital adopts traditional engineering management, while we used the SWOT-STPA combined method to analyze the external environment and internal conditions, as well as the characteristics and technical aspects of the building. In other words, traditional management is rough and initiated by the manager. The method we proposed is meticulous and initiated from the characteristics of the building and the basic technical processes. Lastly, the revised manuscript made modifications to section 2, focusing on the deficiencies in the current research perspective. In section 3, an analysis of the main issues faced by emergency buildings was added, and in section 5, a discussion was added. These changes contribute to future research.

Summary of responses to reviewer 2

I sincerely thank you on behalf of all the authors for your comments. We have taken your feedback into consideration and made revisions and responses accordingly. We hope you can give us a chance to complete our publication on PLOS. If you have any questions, please feel free to ask, and we would be more than happy to communicate with you. Thank you once again for your comments.

---

## [Editor Report · Decision Letter 1]

15 Nov 2023

Strategic management and risk control of emergency hospital construction: SWOT and STPA framework from a systems thinking perspective

PONE-D-23-27781R1

Dear Dr. Zhu,

We’re pleased to inform you that your manuscript has been judged scientifically suitable for publication and will be formally accepted for publication once it meets all outstanding technical requirements.

Kind regards,

Sudipta Chowdhury

Academic Editor

PLOS ONE

Additional Editor Comments (optional):

You have adequately addressed the concerns raised by the reviewers, especially reviewer 2. Given the significant revision made in the original manuscript, this article is deemed appropriate for publication in the journal. Congratulations!
---

## [Editor Report · Acceptance letter]

20 Nov 2023

PONE-D-23-27781R1 

Strategic management and risk control of emergency hospital construction: SWOT and STPA framework from a systems thinking perspective 

Dear Dr. Zhu:

I'm pleased to inform you that your manuscript has been deemed suitable for publication in PLOS ONE. Congratulations! Your manuscript is now with our production department. 

Kind regards, 

on behalf of

Dr. Sudipta Chowdhury 

Academic Editor

PLOS ONE